# Comparative Analysis of Cemented and Cementless Straight-Stem Prostheses in Hip Replacement Surgery for Elderly Patients: A Mid-Term Follow-up Study

Marco Sapienza, Danilo Di Via, Marco Simone Vaccalluzzo, Luciano Costarella, Vito Pavone * and Gianluca Testa

Department of General Surgery and Medical Surgical Specialties, Section of Orthopaedics and Traumatology, P.O. "Policlinico Gaspare Rodolico", University of Catania, 95100 Catania, Italy; marcosapienza09@yahoo.it (M.S.); danilodivia91@gmail.com (D.D.V.); marcovaccalluzzo@hotmail.it (M.S.V.); lcostarella@yahoo.it (L.C.); gianpavel@hotmail.com (G.T.)
* Correspondence: vitopavone@hotmail.com

**Abstract:** This retrospective cohort study assesses the effectiveness of straight-stem cementless versus cemented prostheses in hip replacement surgeries for elderly patients with femoral neck fractures. We analyzed 80 patients aged 70 and over who underwent surgery between 2018 and 2021. Clinical outcomes were evaluated using the Harris Hip Score, WOMAC Score, and Visual Analogue Scale, alongside radiological assessments through Brooker's classification. Preoperative Dorr classification and five postoperative criteria (subsidence, cortical hypertrophy, pedestal sign, radiolucent lines, and stress shielding) were used to assess implant efficacy. The results demonstrated satisfactory mid-term outcomes for both groups, with slightly higher clinical scores observed in the cementless stem group. The Harris Hip Score (HHS) averaged $74.4 \pm 6.7$ in the cemented group and $79.2 \pm 10.4$ in the cementless group, with a statistically significant difference ($p = 0.0146$). The WOMAC Score showed an average of $30.1 \pm 4.6$ in the cemented group compared to $27.1 \pm 6.9$ in the cementless group, also indicating a statistically significant improvement ($p = 0.0231$). However, radiographic findings call for a re-evaluation of long-term stability. Our statistical analysis, which included power calculation and multivariate analysis to adjust for confounding variables, offers a comprehensive assessment of implant effectiveness. The findings contribute to the ongoing debate on the choice between cemented and cementless prostheses, indicating that both are viable options catering to different patient needs. Further research overcoming this study's limitations is crucial for a deeper understanding of optimal treatment strategies in hip replacement surgery for the elderly.

**Keywords:** hip replacement surgery; femoral neck fractures; cemented stem; uncemented stem; clinical outcomes; radiographic outcomes; mid-term follow-up

## 1. Introduction

Femoral neck fractures represent prevalent traumatic injuries that may result in substantial disability, particularly among the elderly [1]. The primary treatment for these fractures is surgical intervention, which includes options such as internal fixation, total hip arthroplasty (THA) [2], and hemiarthroplasty [3]. Hip replacement surgery emerges as the elective choice for treating older adults and the elderly [4]. The decision to proceed with hip replacement surgery for older adults is multifaceted, influenced by the severity of the femoral neck fracture, overall patient health profile, comorbidities, bone quality, patient mobility levels, and life expectancy considerations.

Hip replacement surgery stands as one of the most frequently performed procedures in global orthopedic practices, owing to its consistently favorable outcomes. In Italy, over 90,000 surgeries were documented in 2021 [5], covering emergency procedures for medial femoral neck fractures, elective surgeries (e.g., for osteoarthritis or femoral head necrosis),

and surgical revisions. The selection of the surgical approach (THA vs. hemiarthroplasty), the choice of the prosthetic implant type, and the appropriate stem configuration (straight vs. curved [6], cementless [7] vs. cemented [8,9], etc.) hinge on preoperative assessments, encompassing both clinical and radiological considerations.

To fully comprehend the context within which hip replacement surgeries are performed, it is crucial to acknowledge the regional differences in preference for cemented versus cementless implants. These differences are not merely a matter of clinical outcomes but are deeply influenced by medical system guidelines, availability of technology, surgeon expertise, and cultural preferences towards postoperative rehabilitation. For instance, Scandinavia [8] shows a predominance of cemented implants due to their long-standing tradition and favorable registry outcomes, whereas North America [8] exhibits a stronger inclination towards cementless options, reflective of a broader trend towards rapid mobilization and biological fixation.

Our study aims to directly compare the postoperative outcomes and long-term effectiveness of cemented versus cementless straight-stem prostheses in hip replacement surgery for elderly patients suffering from medial femoral neck fractures. We hypothesize that, while both methods provide satisfactory mid-term outcomes, there will be discernible differences in terms of implant longevity, patient mobility, and complication rates that could inform implant selection based on patient-specific factors.

## 2. Materials and Methods

### 2.1. Study Design and Patient Characteristics

This study was designed as a retrospective comparative cohort study involving 80 patients who underwent hip replacement surgery subsequent to displaced medial femoral neck fractures at the Orthopedic Clinic of Policlinico "G. Rodolico" in Catania. The inclusion period spanned approximately four years, from February 2018 to December 2021. Preoperative radiographic scans were utilized to classify patients' fractures according to Garden's criteria [9], topographic classification [10], and the Dorr classification [11]. The cohort consisted of 44 patients receiving total hip arthroplasty (THA) and 36 treated exclusively with hemiarthroplasty (HA), with the choice between THA and HA based on individual clinical considerations, including bone quality assessment and comorbidities.

Inclusion criteria: Patients admitted from 29 February 2018 to 11 December 2021, aged over 70 years, with traumatic medial (intracapsular) femoral neck fractures (Garden III or IV, displaced fractures) treated exclusively through hemiarthroplasty or total hip arthroplasty using straight stems (both cemented and cementless).

Exclusion criteria: Patients outside the age range of 70 to 90 years, those with non-traumatic fractures, lateral femoral neck fractures (extracapsular), Garden I or II medial femoral neck fractures (nondisplaced), and those treated through internal fixation, hip resurfacing surgery, or with curved stems.

To eliminate confusion regarding the age criteria for our study, the inclusion criteria have been specified as patients aged 70 years and above, reflecting our focus on the elderly population. Conversely, the exclusion criteria have been adjusted to exclude patients aged 90 years and above, to concentrate on a demographic where the benefits of surgery outweigh the operative risks. This adjustment ensures a clear demarcation in the study population, facilitating precise analysis and relevant findings.

Patients were allocated to either the cemented or cementless treatment group based on a combination of clinical judgment and specific criteria including bone quality, patient activity level, and underlying health conditions. This allocation process reflects our commitment to personalized medicine, ensuring that each patient received the treatment option best suited to their individual needs.

### 2.2. Surgical Procedure

The surgical procedure employed for these patients was the modified Watson–Jones technique (anterolateral access, [12]) in all cases, with Müller Straight Stem (Zimmer®,

Winterthur, Switzerland) [12,13] used for cemented prostheses and CLS Straight Stem (Zimmer®, Warsaw, IN, USA) [14] for cementless ones. This selection was based on their established clinical efficacy, innovative design, and compatibility with the needs of elderly patients.

The surgeries were performed by a total of 4 expert surgeons.

The types of bone cement used, and the cementation techniques were chosen based on current best practices and literature guidelines. We utilized the high viscosity, radiopaque bone cement to optimize implant adherence and stability.

### 2.3. Anesthesia Methods

The choice of anesthesia method, between general and spinal, was personalized based on patient preferences and comorbidities after an interview with anesthesiologists that informed patients about the risks of the two anesthesiological procedures.

### 2.4. Clinical and Radiographic Assessment

The results were systematically organized to facilitate a comparative analysis of the two subgroups within the cohort: patients with a cementless straight-stem implant and those with a cemented one.

As per the protocol of the Orthopedic Clinic, a clinical and radiological follow-up at at least 1 month and 1 year post surgery was mandated for all patients, with additional surveillance visits conducted in 2022. Any clinical and radiological evaluations by patients were integrated.

Surveillance visits involved clinical and functional assessments based on the Harris Hip Score (HHS) [15], the WOMAC Score [16], and the Visual Analogue Scale (VAS) [17]. Radiological outcomes were evaluated using Brooker's classification [18] based on radiographic scans performed during the follow-up period.

In this study, we set a threshold of 2 mm for subsidence based on clinical considerations and a review of the existing literature [7,19,20], recognizing it as a potential indicator of compromised implant stability. This threshold was chosen to facilitate early detection of cases requiring closer observation for potential implant failure risks, although it is acknowledged that the clinical implications of subsidence are multifactorial and may vary by patient and implant type.

### 2.5. Statistical Analysis

The cemented and cementless implant groups were statistically compared using Student's *t*-test for quantitative data (average age) and the chi-squared test for qualitative data (Garden classification, fracture localization). HHS, WOMAC, and VAS scores of the two groups at one year were compared using Student's *t*-test; postoperative parameter comparison was performed using the chi-squared test.

To ensure the robustness of our findings, we conducted power calculations to determine the sample size required to detect significant differences between groups with an alpha level of 0.05 and a power of 80% (Figure 1). Multivariate regression analyses were performed to adjust for confounders, with results reported as adjusted odds ratios (aORs) for categorical outcomes and adjusted mean differences (aMDs) for continuous outcomes. Confidence intervals (95% CIs) are provided to reflect the precision of our estimates.

All *p*-values are reported as specific numbers to provide a detailed understanding of the results. We acknowledge that the number of patients and demographic characteristics in Table 1 do not require a *p*-value comparison. The statistical analysis was carefully applied and described, ensuring the appropriateness of tests used for qualitative data comparisons.

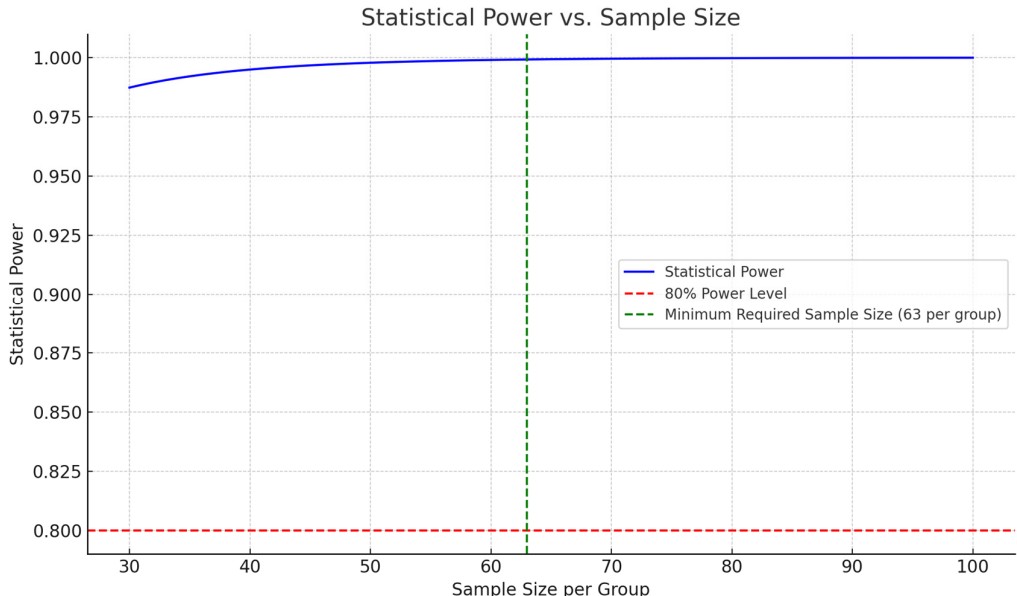

**Figure 1.** Graphical representation of statistical power versus sample size for Harris Hip Score differences. This graph illustrates the relationship between sample size per group and the statistical power of this study. The red dashed line indicates the desired power level of 80%, while the green dashed line marks the minimum required sample size of 63 patients per group to achieve this power, assuming a standard deviation of 10 and an expected mean difference of 5 points in Harris Hip Scores between the cemented and cementless hip replacement groups. The blue line represents the calculated statistical power across a range of sample sizes, demonstrating how increasing the sample size enhances this study's ability to detect significant differences.

**Table 1.** Anthropometric and preoperative characteristics of the study cohort.

| | Total Cohort | Cemented Implant Group | Cementless Implant Group | *p*-Value |
|---|---|---|---|---|
| Number of patients | 80 | 28 | 52 | |
| Average age | 80.4 ± 6-5 | 85.8 ± 3.4 | 77.5 ± 5.8 | 0.2520 |
| Sex | 21 males | 10 males | 11 males | |
| | 59 females | 18 females | 41 females | |
| Garden III | 32.5% | 28.6% | 34.6% | 0.764 |
| Garden IV | 67.5% | 71.4% | 65.4% | 0.764 |
| Transcervical fractures | 17.5% | 14.3% | 19.2% | 0.805 |
| Basicervical fractures | 28.75% | 25% | 30.8% | 0.776 |
| Subcapital fractures | 53.75% | 60.7% | 50% | 0.495 |
| Dorr Type A | 8.75% | 7.1% | 9.6% | 1.0 |
| Dorr Type B | 52.5% | 50% | 53.8% | 0.925 |
| Dorr Type C | 38.75% | 42.9% | 36.6% | 0.754 |

## 3. Results

### 3.1. Demographic Data

From the initial cohort of 94 patients selected for this study, 14 individuals succumbed during the follow-up period and were consequently excluded from the analysis. The surviving 80 patients comprised 21 males (10 in the cemented hip replacement group and 11 in the cementless group) and 59 females (18 in the cemented hip replacement group and 41 in the cementless group). The cohort treated with a cemented implant consisted of 28 subjects, while the cohort treated with a cementless implant included 52 subjects. The mean age of the entire patient population was 80.4 ± 6.5 years, with an average age of 85.8 ± 3.4 years for those subjected to cemented hip replacement and 77.5 ± 5.8 years for those receiving cementless hip replacement. Our comparative analysis demonstrated

no significant differences in age (*p* = 0.2520) and sex distribution (*p* = 1.0) between the cemented and cementless implant groups, confirming the comparability of our cohorts.

We classified the fractures according to the Garden classification and topographical classification. The fracture classifications, including Garden III (*p* = 0.764) and Garden IV (*p* = 0.764), transcervical (*p* = 0.805), basicervical (*p* = 0.776), subcapital fractures (*p* = 0.495), and Dorr classifications (Type A: *p* = 1.0, Type B: *p* = 0.925, Type C: *p* = 0.754), also showed no statistically significant differences, indicating a balanced distribution of fracture types across both groups (Table 1).

### 3.2. Other Surgical Details

The average surgical time was 95 min for cemented prostheses and 87 min for uncemented prostheses. The average blood loss was 300 mL. The mean Charlson Comorbidity Index of the patients was 2, indicating a moderate level of pre-existing comorbidities.

### 3.3. Clinical and Radiographic Assessment and Follow-up

As previously stated, a follow-up was conducted for all patients who underwent hip replacement surgery, spanning an average duration of 36 months. With clinical and X-ray evaluations at 1 month, 3 months, 6 months, 12 months, 18 months, 24 months, and 36 months postoperative. Surveillance visits facilitated the assessment of patients' clinical outcomes using the previously mentioned scores, including the Harris Hip Score (HHS), WOMAC Score, and Visual Analogue Scale (VAS).

Comparative analysis between the cemented and cementless implant groups demonstrated notable differences in clinical outcomes.

The Harris Hip Score (HHS) averaged 74.4 ± 6.7 in the cemented group and 79.2 ± 10.4 in the cementless group, with a statistically significant difference (*p* = 0.0146).

The WOMAC Score showed an average of 30.1 ± 4.6 in the cemented group compared to 27.1 ± 6.9 in the cementless group, also indicating a statistically significant improvement (*p* = 0.0231).

However, the Visual Analogue Scale (VAS) for pain did not show a significant difference between the groups, with scores of 2.3 ± 1 for the cemented and 2 ± 1.3 for the cementless implants (*p* = 0.2547) (Table 2 and Figure 2).

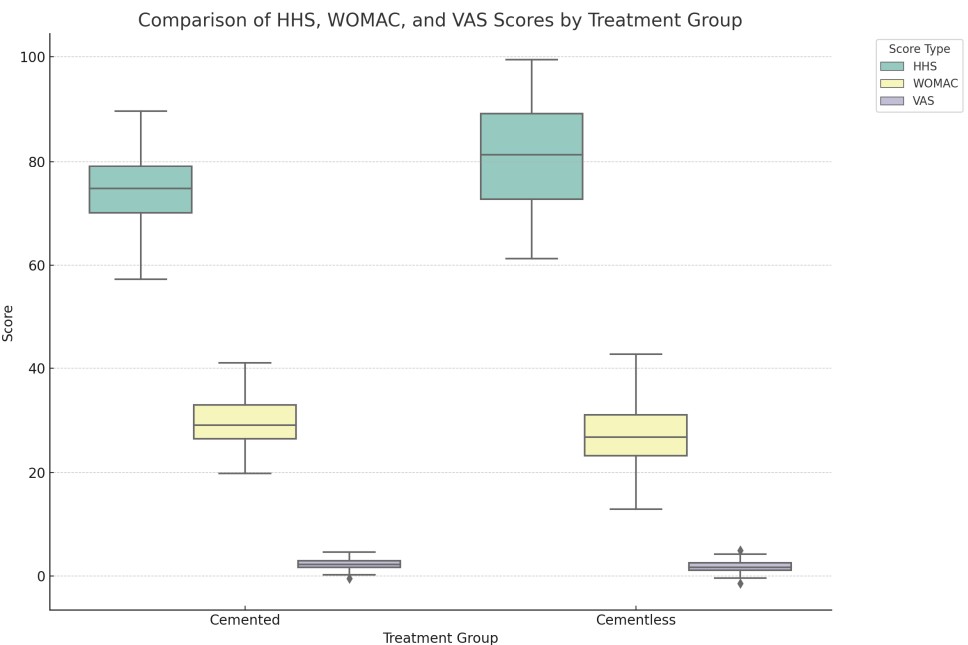

**Figure 2.** Comprehensive box plot illustrating the distribution of HHS, WOMAC, and VAS scores across the cemented and cementless hip replacement groups. Each box plot delineates the median,

interquartile range, and outliers for the scores, showcasing clinical outcomes (HHS), patient-reported outcomes related to pain, stiffness, and physical function (WOMAC), and patient-perceived pain intensity (VAS). This all-encompassing visualization underscores the multifaceted nature of treatment outcomes, facilitating a nuanced comparison across different types of assessments and treatment groups.

**Table 2.** Clinical outcomes of the study cohort.

| | Total Cohort | Cemented Implant Group | Cementless Implant Group | *p*-Value |
|---|---|---|---|---|
| HHS | $77.5 \pm 9.5$ | $74.4 \pm 6.7$ | $79.2 \pm 10.4$ | 0.0146 |
| WOMAC Score | $28.2 \pm 6.4$ | $30.1 \pm 4.6$ | $27.1 \pm 6.9$ | 0.0231 |
| VAS | $2.1 \pm 1.2$ | $2.3 \pm 1$ | $2 \pm 1.3$ | 0.2547 |

After the surveillance follow-up, an extensive analysis was conducted, encompassing patients' preoperative, postoperative, and subsequent radiographic scans. The assessment employed Brooker's classification [20] and five postoperative parameters, which included subsidence [21], cortical hypertrophy [22], pedestal sign [23], radiolucent lines [24], and stress shielding [25]. These criteria served as instrumental tools for evaluating the efficacy of the implanted prosthetic replacement and predicting the potential occurrence of complications.

The evaluation of heterotopic ossification using Brooker's classification indicated a non-significant difference in the incidence of heterotopic ossification among our groups (Brooker 0: $p = 0.144$, Brooker I: $p = 0.980$, Brooker II: $p = 0.546$, Brooker III: $p = 0.237$), underscoring the similarity in postoperative ossification patterns regardless of the implant type used (Table 3).

**Table 3.** Incidence of heterotypic ossification (HO) in the study cohort.

| | Total Cohort | Cemented Implant Group | Cementless Implant Group | *p*-Value |
|---|---|---|---|---|
| Brooker 0 | 45% | 32.1% | 51.9% | 0.144 |
| Brooker I | 33.75% | 35.7% | 32.7% | 0.980 |
| Brooker II | 16.25% | 21.4% | 13.4% | 0.546 |
| Brooker III | 5% | 10.8% | 2% | 0.237 |

Subsidence: This finding is considered negligible if less than 2 mm and pathological if exceeding 2 mm. In the total cohort, negligible subsidence was observed in 71 patients (88.75%), while significant subsidence occurred in 9 patients (11.25%). No statistical differences between the groups ($p > 0.05$) (Table 4).

**Table 4.** Postoperative parameters of the study cohort.

| | Total Cohort | Cemented Implant Group | Cementless Implant Group | *p*-Value |
|---|---|---|---|---|
| Subsidence > 2 mm | 11.25% | 11.7% | 11.5% | 1.0 |
| Pedestal sign | 47.5% | - | 73.1% | - |
| Stress shielding | 48.75% | 25% | 61.5% | 0.0039 |
| Cortical hypertrophy | 52.5% | 35.7% | 69.2% | 0.0079 |
| Radiolucencies >2 mm | 26.25% | 25% | 26.9% | 1.0 |

Pedestal sign: In the total cohort, the absence of the pedestal sign was noted in 42 patients (52.5%), while its presence was observed in 38 patients (47.5%) (Table 4).

Stress shielding: The phenomenon of stress shielding was observed in 39 patients (48.75%) in the total cohort and found absent in 41 patients (51.25%). Stress shielding show-

ing a significant difference with a higher incidence in the non-cemented group (*p* = 0.0039) (Table 4).

Cortical hypertrophy: Cortical hypertrophy was observed in zones 3 and 5 in 42 cases (52.5%) in the study cohort, while it was absent in the remaining 38 cases (47.5%). Statistical differences between the groups (*p* = 0.0079) (Table 4).

Radiolucent lines: Periprosthetic lucencies, if less than 2 mm wide, are considered normal findings. Lucencies that are wider than 2 mm and/or progressive are indicative of abnormality. The total cohort exhibited abnormal radiolucent lines in 21 patients (26.25%), while normal findings were observed in the remaining 59 patients (73.75%). No statistical differences between the groups (*p* >0.05) (Table 4).

We further explored the longevity of the prostheses using Kaplan–Meier survival analysis, comparing survival curves with the log-rank test. Although no statistically significant differences were observed (*p* > 0.05), a trend towards lower revision rates was noted for the cementless group.

## 4. Discussion

Our findings underscore the importance of considering patient-specific factors such as bone quality and comorbidities when selecting the type of prosthetic implant. The slight superiority of cementless stems observed in our study aligns with recent trends favor their use in younger, more active patients due to their potential for biological fixation and long-term stability. However, the choice between cemented and cementless prostheses should be guided by a comprehensive assessment of each patient's clinical and radiological profile.

The selection of the optimal fixation method should be driven by clinical outcomes, particularly implant survivorship [26]. Cemented fixation, while cost-effective, requires extended surgical time and is associated with complications such as cement aging, microfractures, or late loosening, especially in younger patients [27]. Additionally, cement implantation syndrome is a significant concern with potential life-threatening consequences, especially in patients with cardiovascular comorbidities [28]. On the contrary, cementless fixation is quicker to perform but is associated with higher costs and complications such as thigh pain and stress shielding [29].

While the literature suggests that both cemented and cementless femoral components exhibit excellent outcomes when the procedure is performed to a high standard, there is insufficient evidence to support the notion that patients with poor outcomes with cementless implants would necessarily experience better outcomes by converting to cemented implants. Moreover, although certain cemented femoral components have demonstrated a good long-term track record, the same cannot be asserted for cemented acetabular components.

Studies reporting excellent long-term survivorship for both cemented [30] and cementless [31,32] stems have faced challenges due to factors such as small sample sizes, short follow-up durations, or retrospective study designs, which may not accurately represent regional or national composite data [31]. Despite the existence of randomized controlled trials (RCTs), biases and study limitations persist, with some studies indicating that RCTs are more prone to heterogeneity in results than observational studies [33,34].

Zhang et al. [35] conducted a review based on evidence from international joint registries, randomized clinical trials, and meta-analyses. They concluded that cemented fixation demonstrated overall better long-term survivorship than cementless fixation in primary hip replacement. Although, many factors and possible complications must be considered when implanting a cemented hip arthroplasty [35], cemented fixation notably performed better in older patients, while cementless fixation showed better outcomes in younger patients. However, the study's limitation lies in its reliance on large databases, with physiological age and activity profiles not considered, using only chronological age for analysis. Survivorship was evaluated as an outcome rather than assessing the quality of life, leading to potential oversights, such as painful cemented stems in elderly patients

that cannot be revised but still require increased community care, impacting overall hip prosthesis costs.

In summary, the multifaceted nature of outcomes in hip replacement surgery underscores the importance of carefully considering various patient factors, such as bone quality, medical comorbidities, and surgeon expertise, rather than solely relying on patient age. The overarching objective of arthroplasty is to improve individual patient outcomes, leading to tangible benefits in both health and economic aspects for society.

Our study found satisfactory clinical and radiological outcomes in the mid-term follow-up for all patients in the cohort, with the group treated with cementless stem implants showing slightly superior results compared to the group treated with cemented stem implants (Tables 2–4).

The inclusion of PROMs, such as the WOMAC Score and VAS, in our study highlights the critical role of patient-reported outcomes in evaluating the success of hip replacement surgery. These measures provide invaluable insights into patient perceptions of pain, functionality, and overall post-surgery satisfaction, emphasizing the need for a patient-centered approach in orthopedic care. Future research should delve deeper into the correlation between clinical measures and PROMs to optimize postoperative management strategies.

The statistically significant differences observed in both the Harris Hip Score ($p = 0.0146$) and the WOMAC Score ($p = 0.0231$) suggest a superior clinical outcome for patients receiving cementless implants, aligning with the hypothesis that cementless options might offer better long-term stability and patient satisfaction. This finding is consistent with recent literature suggesting the potential advantages of cementless implants in specific patient demographics. However, the lack of significant difference in VAS scores ($p = 0.2547$) indicates that pain perception post surgery may be influenced by factors beyond the choice of cemented versus cementless implant types. These results underscore the importance of considering a multifaceted approach to patient care, where both clinical outcomes and patient-reported outcomes are valued. Further research is required to explore the nuances of these findings and their implications for surgical practice.

Although many of our comparisons showed *p*-values greater than 0.05, indicating a lack of statistically significant difference between groups, we acknowledge this does not preclude clinically relevant differences. These findings might reflect the limited statistical power due to the sample size and prompt the need for larger studies to further explore these trends.

Brooker's classification revealed that most of the cohort showed no significant heterotopic ossification (Brooker 0 45%, Brooker I 33.75%). The group treated with cementless implants demonstrated a higher percentage of Brooker 0 than the other group (51.9% vs. 32.1%). The incidence of significant heterotopic ossification (Brooker II and III) was slightly lower in the cementless implant group. The results showed a relatively homogeneous distribution of bone quality in both groups, with the cementless implant group having slightly higher percentages of Dorr Types A and B. Radiologically relevant subsidence (>2 mm) was found in approximately 1/10 of patients in all groups, indicating a potential need for revision surgery. Relevant percentages of stress shielding, cortical hypertrophy, abnormal radiolucent lines, and the pedestal sign were observed in the total cohort, with some differences between the two groups (Table 4).

In light of varying bone densities across different patient demographics, implant stability, and outcomes, the requirements for bone grafting during hip replacement surgery merit discussion. Reference [36] outlines the potential for bone grafting to mitigate the challenges posed by diminished bone quality, offering a pathway to enhance implant stability and longevity.

## 5. Study Limitations

The present study acknowledges certain limitations that warrant consideration. Notably, the patient cohort exhibits significant demographic variations, including differences in age, sex, and specific fracture types. These disparities may introduce confounding vari-

ables that could impact the generalizability of this study's findings to broader populations. This investigation was restricted to patients who received straight-stem implants, with the exclusion of those treated with curved stems, and we did not differentiate by additional populations of patients who received a THA and HA to avoid confusing the reader. This inevitably limits the extrapolation of results to the broader spectrum of hip implant systems.

An important limitation lies in the absence of comprehensive anthropometric data, such as Body Mass Index (BMI), and anamnestic information, including the presence of osteoporosis, diabetes, musculoskeletal diseases, and other comorbidities.

Moreover, this study describes a mid-term follow-up. Certain radiological outcomes, such as abnormal radiolucent lines, necessitate ongoing evaluation to determine their long-term significance.

This study may be susceptible to selection bias due to the inadequate consideration of certain factors, such as bone quality, during patient selection. This potential bias could influence the observed outcomes and limit the applicability of this study's conclusions.

The decision to compare procedures based on patient age rather than considering other critical factors, such as bone quality or comorbidities, may introduce a limitation.

In summary, while this study contributes valuable insights into medial femoral neck fractures and their treatment, these acknowledged limitations emphasize the need for cautious interpretation of the results. Future research endeavors should aim to address these limitations for a more comprehensive understanding of the complex dynamics associated with hip replacement surgeries.

### 6. Conclusions

This study has demonstrated proficient surgical and clinical performance, yielding satisfactory results in both groups of patients within the total study cohort. This underscores the appropriateness of stem selection in most cases, aligning with the patients' preoperative health status and quality of life. Moreover, at mid-term follow-up, patients with cementless stem implants showed higher complication rates than patients with cemented prostheses. To provide more definitive insights into the effectiveness of cementless stems compared to cemented stems in hip replacement surgery, further studies involving larger cohorts are imperative.

**Author Contributions:** Conceptualization, M.S., D.D.V. and G.T.; methodology, V.P.; software, M.S.V.; validation, M.S., V.P. and L.C.; formal analysis, L.C.; investigation, M.S.V.; resources, M.S.; data curation, L.C.; writing—original draft preparation, M.S.V.; writing—review and editing, M.S.; visualization, L.C.; supervision, V.P.; project administration, G.T.; funding acquisition, V.P. All authors have read and agreed to the published version of the manuscript.

**Funding:** This research received no external funding.

**Institutional Review Board Statement:** This study was conducted in accordance with the Declaration of Helsinki.

**Informed Consent Statement:** Informed consent was obtained from all subjects involved in this study.

**Data Availability Statement:** The data presented in this study are available in the article.

**Conflicts of Interest:** The authors declare no conflicts of interest.

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
