# Peer review of "Comparative Analysis of Cemented and Cementless Straight-Stem Prostheses in Hip Replacement Surgery for Elderly Patients: A Mid-Term Follow-up Study"

_prosthesis, doi:10.3390/prosthesis6030038_

Round 1

Reviewer 1 Report

Comments and Suggestions for Authors

In the introduction section:     

- consider expanding the discussion of factors influencing surgical decisions to include: patient overall health profile, comorbidities, bone quality

- how are regional preferences affected previous outcomes in other studies?

In the discussion section:

- "recent trends favoring" should be "recent trends favor" - please reread the manuscript in order to address all English inconsistencies

- how does severity of comorbidities or prior surgical history influence the choice between cemented and cementless options?

- integrate the discussion on bone grafting with the main narrative on implant stability and outcomes

Results section is correctly presented

In the methods section:

- justify the age cut-off

- describe more on the criteria used to choose between anesthesia types

- the study lacks a clear description of the randomization process

Conclusions confirm the hypothesis

References are adequate

Title is ok

Comments on the Quality of English Language

  • "recent trends favoring" should be "recent trends favor" - please reread the manuscript in order to address all English inconsistencies

Author Response

Reviewer 1:

Dear reviewer,

Thank you for your important contribution, you will find the required revisions in the text, and you can identify them through track changes.

In the introduction section: 

- We included: the patient overall health profile, comorbidities, bone quality

- Because using only cemented or non-cemented prostheses their results do not compare the results of the two techniques.

In the discussion section:

- I corrected English language errors.

- I have explained what is required from line 301.

In the methods section:

- I have specified in the methods section the reason for the choice of age cut-off, you will find it underlined in yellow in the text.

- The patient after the interview with the anesthesiologist chooses the type of anesthesia, after being informed of the risks associated with the procedures and after an overall assessment of the patient’s health.

- No randomization was performed due to the retrospective nature of the study and homogeneous data were used.

I reviewed the English language as requested.

Reviewer 2 Report

Comments and Suggestions for Authors

1. Abstract: In the results section of the abstract there should be some real results with statistics and numbers. Without numbers it is more like a short discussion. 

2. Lines 30-33: This is where you should cite this meta-analysis (DOI: 10.1186/s13018-023-04114-8) as it is based on the largest cohort of patients studied on this topic. 

3. Line 39: Do you refer to the present study? I would leave this sentence here and move it to the limitations section. 

4. Line 65: The methods section lacks institutional ethical approval. If this has not been obtained, you should apply for it retrospectively. The study design should follow the STROBE guidelines. 

https://www.equator-network.org/?post_type=eq_guidelines&eq_guidelines_study_design=observational-studies&eq_guidelines_clinical_specialty=0&eq_guidelines_report_section=0&s=+&eq_guidelines_study_design_sub_cat=0  

5. Line 179: Table 1. We are scientists and we know that there are only two human sexes - male and female. The concept of gender is a product of politicians. Change "gender" to "sex". The bold print below the p-value should only be used where there is statistical significance. 

6. Line 200: Table 2. "Total cohort", "Cemented implant group" and "Cementless implant group" should be reported exactly as in Table 1. In Table 2 there are unnecessary hyphens and a missing space between "implant" and "group" ("implantgroup"). Again, the bold print below the p-value should only be used where there is statistical significance. 

7. Line 222: Table 3. Again, an unnecessary hyphen in "Total co-hort". Again, the bold print below the p-value should only be used where there is statistical significance. 

8. Line 247: Table 4. Again, an unnecessary hyphenin "Total co-hort". Here, bold below the p-value is ok.

9. Line 249: You should present the main findings of the study first. 

10. Lines 336-381: The limitations section is far too long and should be shortened.  

11. Lines 382-399: The conclusion section is far too long and should be shortened. A conclusion section is usually 2-3 sentences long. 

Author Response

Reviewer 2:

Dear reviewer,

Thank you for your important contribution, you will find the required revisions in the text, and you can identify them through track changes.

  1. In the abstract results section, I added real results with statistics and numbers.
  2. Thanks for the suggested reference. I added meta-analysis (DOI: 10.1186/s13018-023-04114-8) as a second bibliographic reference.
  3. I deleted the sentence from the introduction as required and moved part of it within the study limits.
  4. No ethical committee approval is expected from our department in retrospective studies. Also in the link attached by you is not expected approval of the ethics committee for retrospective studies.
  5. I Change "gender" to "sex" in all parts of the text.
  6. I modified Tables 2, 3 and 4 as you requested.
  7. I modified Tables 2, 3 and 4 as you requested.
  8. I modified Tables 2, 3 and 4 as you requested.
  9. I presented the main findings at the beginning of the study as you requested.
  10. I have shortened the limitations of the study section, as you suggested.
  11. I have shortened the conclusions of the study with the most meaningful and appropriate sentences for this section, as suggested by you.

Reviewer 3 Report

Comments and Suggestions for Authors

Thank you for the opportunity to review this manuscript. In this study, the authors evaluated the clinical and radiological outcomes of 80 (they should state this number instead of 94) patients treated with hip arthroplasty for femoral neck fractures. They compared outcomes between cemented and cementless femoral stems. The authors reported that, while mid-term outcomes were adequate for both groups, clinical scores were slightly superior in the cementless stem group. I commend the authors for conducting this study, however, some important concerns remain:

Major comments:

-        My major concern is the combination of THA and HA in all statistical analysis. The authors state that 54 patients had THA and 40 had HA, however, out of the final 80 patients analyzed, how many had THAs and how many HA? This could introduce important confounding in the comparison of cementless vs. cemented. I would recommend conducting separate analysis of cemented vs. cementless for the THA cohort, and for the HA cohort.

-        The real sample size is 80 patients, not 94 (as 14 patients were excluded initially)

-        What is the follow-up of each treatment group?

-        Limitations should be described narratively instead of using headings.

-        First paragraph of the conclusions is not a real conclusion statement.

Minor comments:

-        Line 55: Please remove the word “Canada”, it should be North America alone (includes US, Mexico and Canada).

-        Line 78: Please rephrase to “Treated exclusively with hemiarthroplasty…”

-        Lines 87-88: Please rephrase to “where the benefits of surgery outweigh the operative risks”

-        Lines 108-110: information on surgical time and average blood loss should be displayed on the Results section.

Comments on the Quality of English Language

Adequate.

Author Response

Reviewer 3:

Dear reviewer,

Thank you for your important contribution, you will find the required revisions in the text, and you can identify them through track changes.

Major comments:

  • I’ve edited sample size in 80 patients throughout the study, including the abstract. However, I left the number 94 only in the initial part of the results for the record. We have not further divided the two populations (THA and HA) because the study compares cemented and cementless femoral stems. Therefore, we do not want to confuse the reader by adding an additional subdivision. However, I have added within the limits of the study the lack of this further subdivision.
  • I’ve edited sample size in 80 patients throughout the study.
  • I have further specified the type of follow-up performed in paragraph 3.3 of the results (Clinical evaluation and radiography and follow-up).
  • I described the limits narratively without using headers, as required.
  • I deleted the first paragraph of the conclusions, to describe the useful information.

Minor comments:

  • Line 55: I removed the word “Canada”.
  • Line 78: I rephrased to “Treated exclusively with hemiarthroplasty…”.
  • Lines 87-88: I rephrased to “where the benefits of surgery outweigh the operative risks”.
  • I moved information on surgical time and average blood in the Results section.